# hMRP8-ATTAC Mice: A New Model for Conditional and Reversible Neutrophil Ablation

**DOI:** 10.3390/cells11152346

**Published:** 2022-07-30

**Authors:** Danique E. M. Duits, Camilla Salvagno, Elisabeth A. M. Raeven, Kim Vrijland, Marjolein C. Stip, Cheei-Sing Hau, Daphne Kaldenbach, Karin E. de Visser

**Affiliations:** 1Division of Tumor Biology & Immunology, Oncode Institute, The Netherlands Cancer Institute, 1066 CX Amsterdam, The Netherlands; d.duits@nki.nl (D.E.M.D.); cas4005@med.cornell.edu (C.S.); eamraeven@gmail.com (E.A.M.R.); k.vrijland@nki.nl (K.V.); m.c.stip@umcutrecht.nl (M.C.S.); t.hau@nki.nl (C.-S.H.); d.kaldenbach@nki.nl (D.K.); 2Department of Immunology, Leiden University Medical Center, 2333 ZA Leiden, The Netherlands; 3Department of Obstetrics and Gynecology, Weill Cornell Medicine, New York, NY 10065, USA; 4Sandra and Edward Meyer Cancer Center, Weill Cornell Medicine, New York, NY 10065, USA; 5Immunotherapy Laboratory, Laboratory for Translational Immunology, University Medical Center Utrecht, 3584 CX Utrecht, The Netherlands

**Keywords:** neutrophils, neutrophil depletion, novel transgenic mouse model, in vivo neutrophil targeting, cancer

## Abstract

Neutrophils are not only crucial immune cells for the neutralization of pathogens during infections, but they are also key players in tissue repair and cancer. Several methods are available to investigate the in vivo role of neutrophils in these conditions, including the depletion of neutrophils with neutralizing antibodies against Ly6G, or the blockade of neutrophil recruitment with CXCR2 inhibitors. A limited number of transgenic mouse models were generated that rely on the disruption of genes important for neutrophil development or on the injection of diphtheria toxin to induce neutrophil ablation. However, these methods have various limitations, including a lack of neutrophil specificity, a lack of long-term efficacy, or a lack of the ability to conditionally deplete neutrophils. Therefore, we generated a transgenic mouse model for the inducible and reversible ablation of neutrophils using the ATTAC (Apoptosis Through Targeted Activation of Caspase 8) approach. With the ATTAC strategy, which relies on the expression of the caspase 8-FKBP fusion protein, apoptosis is induced upon administration of a chemical dimerizer (FK506 analogue) that facilitates the dimerization and activation of caspase 8. In order to achieve specific neutrophil depletion, we cloned the ATTAC construct under the human migration inhibitory factor-related protein 8 (hMRP8) promotor. The newly generated hMRP8-ATTAC mice expressed high levels of the transgene in neutrophils, and, as a consequence, dimerizer injection induced an efficient reduction of neutrophil levels in all the organs analyzed under homeostatic conditions. In situations with extensive pressure on the bone marrow to mobilize neutrophils, for instance in the context of cancer, effective neutrophil depletion in this model requires further optimization. In conclusion, we here describe the generation and characterization of a new transgenic model for conditional neutrophil ablation and highlight the need to improve the ATTAC strategy for the depletion of large numbers of rapidly generated short-lived cells, such as neutrophils.

## 1. Introduction

Neutrophils are multifaceted innate immune cells that play a major role in the first line of immune defense. Being the first cells to arrive at sites of inflammation, neutrophils can clear pathogens via several routes, including phagocytosis, the release of mediators from their granules, neutrophil extracellular traps (NETs), and antibody-dependent cellular cytotoxicity (ADCC) [1,2]. On top of their well-established role in infection, neutrophils are also a critical component of tumor initiation, tumor growth, and metastasis formation. Neutrophils are constantly produced in the bone marrow under the control of the granulocyte colony stimulating factor (G-CSF) and released in the circulation as terminally differentiated and mature cells [3]. The retention and the release of neutrophils in and from the bone marrow is strictly controlled by the expression of the chemokine receptors CXCR4 and CXCR2, respectively. Both in conditions of infection and cancer, the increased production of G-CSF and CXCR2 ligands results in enhanced neutrophil differentiation in the bone marrow and increased neutrophil release into the circulation [3,4].

To investigate the functional role of neutrophils in homeostasis, infections, and cancer, there is a need for methods and mouse models that allow specific and conditional neutrophil targeting. The current in vivo methods to study neutrophils involve either the depletion of the cells or the blockade of their recruitment. In mice, neutrophils, defined as myeloid cells expressing Ly6C^int^Ly6G^high^, can be depleted using anti-Gr-1 antibodies, which recognize a shared epitope between Ly6C and Ly6G surface markers [5,6]. However, because inflammatory monocytes express high levels of Ly6C, this strategy does not only target neutrophils [5]. A more specific approach to deplete neutrophils is using the anti-Ly6G antibody (clone 1A8) [5]. However, the depletion efficiency of this antibody is strongly reduced after two weeks of treatment, likely due to a neutralizing antibody response against the rat IgG2a antibody. Furthermore, the neutrophil depletion efficacy of the 1A8 antibody depends on the genetic background of the mice, with poor depletion efficacy in C57BL/6J mice [7]. This study also showed that the remaining neutrophils after anti-Ly6G treatment are newly generated cells from the bone marrow with reduced Ly6G expression levels, which might explain their lower depletion efficiency [7]. The authors designed a double antibody-based targeting approach using a combination of anti-Ly6G and anti-rat antibodies that improved neutrophil elimination [7]. Another strategy is to impair neutrophil trafficking using CXCR2-deficient mice or CXCR2 antagonists [8,9]. However, it should be considered that CXCR2 expression is not exclusive to neutrophils, but it is also expressed by other cells, including some cancer cells [10]. Besides antibody- and antagonist-based neutrophil targeting methods, transgenic mouse models can be used to conditionally deplete neutrophils. In the *hMRP8cre;ROSA-iDTR^KI^* mouse model, the Cre-inducible diphtheria toxin receptor (DTR) is mainly present in neutrophils, since Cre-recombinase is expressed under the control of the neutrophil-associated human migration inhibitory factor-related protein 8 (hMRP8) promoter, also known as the neutrophil-associated S100 Calcium Binding Protein A8 (S100A8) promoter [11]. As a result, selective and inducible neutrophil depletion can be achieved by injection of diphtheria toxin (DT). However, also in DTR models, it is known that the prolonged treatment of DT leads to immunogenic responses against DT, lowering the depletion duration [12,13].

In order to prevent immunogenicity while sustaining effective neutrophil depletion, we generated a novel transgenic mouse model for the conditional and reversible ablation of neutrophils using the Apoptosis Through Targeted Activation of Caspase 8 (ATTAC) approach developed by Pajvani et al. [14]. The ATTAC strategy was previously used to successfully and over prolonged periods ablate adipocytes, podocytes, pancreatic β-cells, p16^Ink4a^-positive senescent cells, and cardiomyocytes by inducing apoptosis in these cells [14,15,16,17,18]. In the current study, the hMRP8 promotor was used to drive the expression of the ATTAC transgene. Here, we described and characterized the hMRP8-ATTAC mouse in homeostatic conditions and in a cancer setting. 

## 2. Materials and Methods

### 2.1. Generation of hMRP8-ATTAC Transgenic Mice

hMRP8-ATTAC mice were generated in the animal laboratory facility of the Netherlands Cancer Institute by the Animal Modeling Facility. The hMRP8-ATTAC transgenic construct was designed as follows. The FKBP-Caspase8 fragment was subcloned from the POD-ATTAC construct [16] (kindly provided by P. Scherer, the University of Texas Southwestern Medical Center, Dallas, TX, USA) and inserted into a pRRL2 vector containing the humanMRP8 (hMRP8) promoter and hMRP8 3′ untranslated region [19] (a gift from M. Mazzone, VIB-KU, Leuven, Belgium). An IRES-EGFP fragment was inserted at 3′ of the ATTAC construct. The fragment containing the hMRP8 promoter, FKBP-Caspase8 fusion protein, and the 3′ untranslated region was released by ClaI and PspXI digestion, purified from agarose gel by electroelution and microinjected in the pronucleus of FVB/N zygotes. Progenies were screened for GFP by PCR on toe clip-derived genomic DNA (forward: 5′- CTGGACGGCGACGTAAACGGC-3′; reverse: 5′-TCCTTGAAGAAGATGGTGCG-3′). A total of 9 founders were obtained carrying the transgene, and 3 of them showed GFP expression in circulating neutrophils, as assessed by flow cytometry.

Heterozygous (HET) hMRP8-ATTAC mice were maintained in breeding with wildtype (WT) FVB/N mice, obtained from Janvier, to obtain HET and WT mice. Offspring were genotyped by lysing toe clips from the mice in Direct PCR tail (Viagen) containing 1% proteinase K (Sigma Aldrich, Burlington, MA, USA) followed by a PCR for the hMRP8 promoter (forward: 5′-CACCACAGTCTTCAAGGTTG-3′; reverse: 5′-GGCCATACATCCCTGAAACTGA-3′). After crossing HET mice to obtain homozygous (HOM) mice, the colony was maintained by breeding two HOM mice. Genotyping was performed by quantitative RT-PCR for the *Egfp* gene on toe clip lysate. The TaqMan Copy Number Reference Assay for the mouse *Tfrc* gene (Thermo Fisher, Waltham, MA, USA) was used as the endogenous reference gene. This reference, DNA (5 ng/µL), GFP-specific primers (forward: 5′-GAAGCGCGATCACATGGT-3′; reverse: 5′-CCATGCCGAGAGTGATCC-3′) (Invitrogen, Waltham, MA, USA), and GFP probe (UPL 67, Roche, Basel, Switzerland) were added to TaqMan Universal Mastermix (Thermo Fischer Scientific, Waltham, MA, USA), and the RT-PCR was run on a StepOnePLus PCR system. The copy number of the construct in the mice was calculated using the comparative threshold cycle (CT) method, using a sample containing a single GFP copy as a reference.

### 2.2. In Vivo Experiments

The chemical dimerizer AP20187 was diluted in 100% ethanol according to the manufacturer’s recommendation (Takara Bio Inc., San Jose, CA, USA). Stock solutions of dimerizer (in 100% ethanol) or vehicle (100% ethanol) were further diluted in water containing 2% Tween and 10% PEG-400 (Sigma Aldrich, Burlington, MA, USA), which was injected daily intraperitoneal at a dose of 0.5 μg/g body weight for all experiments. Short-term dimerizer treatment of one week was performed in homozygous female hMRP8-ATTAC mice (age 17–19 weeks old) and age-matched female FVB/N mice. Blood was collected before the start of the treatment and after two days by tail vein puncture. Long-term neutrophil depletion was performed by treating homozygous female hMRP8-ATTAC mice (age 17–23 weeks old) for five weeks. Blood was collected before the start of treatment and after every week of treatment. The age of the mice was based on the average age that our cancer models had when a tumor developed, since we aimed to use the hMRP8-ATTAC mice in a cancer setting.

For the orthotopic transplantation of tumor fragments, donor tumors from *K14-cre;Cdh1^F/F^;Trp53^F/F^* (KEP) mice [20] were collected in ice-cold PBS, were cut in small pieces, and were resuspended in DMEM containing 30% FCS and 10% dimethyl sulfoxide and stored at −150 °C. At the day of transplantation, KEP tumor pieces were thawed, and the tumor pieces (1 mm^2^) were orthotopically transplanted into the mammary fat pad of 8–13 weeks old homozygous female hMRP8-ATTAC mice. Treatment with vehicle, dimerizer, or anti-Ly6G (single loading dose of 400 μg followed by 100 μg three times a week, clone 1A8, BioXCell, Lebanon, NH, USA) started at a tumor size of 25 mm^2^ and continued until the tumor reached 225 mm^2^, when the mice were sacrificed for flow cytometry analysis. For intramammary cell line injections, *Wap-cre;Cdh1^F/F^;Akt1^E17K^* (WEA) breast cancer cells were isolated from donor tumors as described before [21] and cultured in Dulbecco’s Modified Eagle’s Medium (DMEM) supplemented with 8% FCS, 100 IU mL^−1^ penicillin, 100 mg mL^−1^ streptomycin, and 2 mM L-glutamine. Equal cell numbers (5 × 10^5^ cells) were orthotopically injected into the mammary fat pad of homozygous female hMRP8-ATTAC mice. When tumors reached a size of 25 mm^2^, mice were allocated into the vehicle or dimerizer treatment arm and treated daily until the endpoint of the experiment, defined as a tumor size of 225 mm^2^. For both tumor models, mice were palpated twice a week, and the perpendicular tumor diameters of mammary tumors were measured twice a week using a caliper. Blood was collected before the start of the treatment and at a tumor size of 100 mm^2^ by tail vein puncture.

Mice were kept in individually ventilated cages and handled in flow cabinets at the animal laboratory facility of the Netherlands Cancer Institute to minimize the risk of infectious complications. Food and water were provided ad libitum. Animal experiments were approved by the Animal Ethics Committee of the Netherlands Cancer Institute and performed in accordance with institutional, national, and European guidelines for Animal Care and Use.

### 2.3. Southern Blot

Genomic DNA was isolated from tissue by proteinase K lysis and organic extraction with phenol-chloroform. Southern blot analysis was performed using 10 µg of genomic DNA, digested with EcoRV to determine the status of transgene integration. Blotting and hybridization was performed as described previously [22]. Primers used for amplification of the GFP probe are the following: forward primer 5′-CCA TGG TGA GCA AGG GCG AGG AGC TG-3′; and reverse primer 5′-CCT TGT ACA GCT CGT CCA TGC CGA GA-3′. The GFP probe was radioactively labeled using the Random primers DNA labeling kit (Thermo Fisher Scientific, Waltham, MA, USA). Hybridization of the GFP probe to EcoRV-digested DNA resulted in a concatemer band of 6114 kb.

### 2.4. Flow Cytometry

Tissues were collected in ice-cold PBS, and blood was collected in heparin or in EDTA-coated tubes (for tail vein puncture at time points). Tumors were mechanically chopped using the McIlwain Tissue Chopper (Mickle Laboratory Engineering, Gomshall, UK) as described [23], and spleen, liver, and lungs were minced using a scalpel. Subsequently, tumor, spleen, and liver were digested in serum-free DMEM containing 3 mg/mL of Collagenase A (Roche) and 25 µg/mL of DNAse (Sigma Aldrich) for 30 min (spleen and liver) or 1 h (tumor) at 37 °C. Lungs were digested in serum-free DMEM containing 100 µg/mL of TM Liberase (Roche) and 25 µg/mL of DNAse for 30 min at 37 °C. Digestion was inactivated by adding DMEM containing 8% FCS, 100 IU mL^−1^ of penicillin and 100 mg mL^−1^ of streptomycin. Samples were then filtered through a 100-μm cell strainer. Lymph nodes were cut manually, digested in serum-free RPMI containing 100 µg/mL TL Liberase (Roche) and 25 µg/mL DNAse for 30 min at 37 °C, and subsequently inactivated in RPMI containing 8% FCS, 100 IU mL^−1^ of penicillin, 100 mg mL^−1^ of streptomycin and β-mercaptoethanol and filtered through a 100-µm cell strainer. Bone marrow cells were flushed out from femora and tibiae and filtered through a 100-µm cell strainer. Erythrocytes were lysed using a NH4Cl buffer (155 mM NH4Cl, 10 mM KHCO3, 0.1 mM EDTA) on the lungs, spleen, liver, blood, and bone marrow. Cells were then washed using FACS buffer (PBS containing 0.5% BSA and 2 mM EDTA) and counted with TC20 Automated Cell Counter (Bio-rad). Equal cell counts (3 × 10^6^) of the single cell suspensions were seeded in a 96-well plat and stained with antibodies for flow cytometry analysis. All cells, except bone marrow cells, were incubated with Fc block (1:50, BD Biosciences, clone 2.4G2) in FACS buffer for 5 min at 4 °C. Cells were then stained with fluorochrome-conjugated antibodies (Appendix A) in FACS buffer for 20 min at 4 °C in the dark. Dead cells were excluded using 7AAD (1:20; eBioscience). Acquisition was performed on a BD LSRII flow cytometer using Diva Software (BD Biosciences), and data analysis was performed using FlowJo software version 9.9.6. Absolute neutrophil counts were calculated for whole blood, one lung, and one femur-tibia as follows: (initial total cell number * frequency of total live neutrophils)/100.

### 2.5. Immunohistochemistry

All immunohistochemical analyses were performed by the Animal Pathology Facility at the Netherlands Cancer Institute. Tissues were fixed for 24 h in 10% neutral buffered formalin and embedded in paraffin. Paraffin sections of 5 μm were cut and deparaffinized, and antigen retrieval was performed with Proteinase K. Samples were then stained with anti-Ly6G antibody (clone 1A8, BD Biosciences, 1:150) or CD31 (polyclonal, Abcam, 1:200). The quantification of positive cells was performed manually by counting five high-power (40x) fields of view (FOV) per tumor. Samples were visualized with a BX43 upright microscope (Olympus), and images were acquired in bright field using cellSens Entry software (Olympus).

### 2.6. Statistical Analysis

Statistical analyses were performed using GraphPad Prism 9 (GraphPad Software Inc., San Diego, CA, USA). The statistical tests that were used are indicated in the figure legends. *p*-values < 0.05 were considered statistically significant.

## 3. Results

### 3.1. Generation and Characterization of the hMRP8-ATTAC Mouse Model

In order to set up a system allowing the genetic depletion of neutrophils in vivo, we utilized the ATTAC approach, in which a mutated FK506-binding-protein (FKBP) domain is fused to caspase 8 [14] and expressed under control of the hMRP8 promoter. The hMRP8 promoter is a myeloid cell-specific promoter that is most abundantly expressed by neutrophils, although some promoter activity can be found in monocytes [24]. Injection of the dimerizer AP20187 induces dimerization of two FKBP domains, leading to activation of membrane-bound myristoylated caspase 8, which should then induce apoptosis of hMRP8-expressing cells. An IRES-EGFP fragment was cloned after the ATTAC construct to allow the detection of transgene positive cells (Figure 1A). The purified transgene fragment was injected in the pronucleus of FVB/N zygotes, and progeny was screened for the presence of GFP. A total of nine transgenic founders carrying the transgene were obtained, of which three founders showed GFP expression in circulating neutrophils with varying intensity (Figure 1B,C). Because founder 3 had the highest GFP expression in neutrophils, we generated the hMRP8-ATTAC line from this founder mouse. The southern blot analysis of founder 3 and heterozygous progeny showed that the construct integrated in the genome in one locus as a concatemer of multiple copies (Appendix A). Circulating neutrophils showed GFP expression in both heterozygous (HET) and homozygous (HOM) hMRP8-ATTAC mice. In line with the double number of transgene copies, neutrophils in HOM mice displayed a more intense GFP expression (Figure 1D), which let us decide to continue our experiments with HOM hMRP8-ATTAC mice. Flow cytometry analyses of several organs of HOM mice showed no differences in immune cell proportion compared to wild type (WT) mice (Appendix A), indicating that the construct does not interfere with immune cell development.

Since S100A8 is not only expressed in neutrophils but also in monocytes and other granulocytes [25], we set out to assess the transgenic GFP expression pattern across different immune cell types in hMRP8-ATTAC mice by flow cytometry analysis. Neutrophils showed GFP expression in all the organs analyzed (i.e., blood, spleen, lung, liver, and lymph node) of hMRP8-ATTAC mice (Figure 1E–I and Appendix A). Further analyses of other myeloid cell populations showed elevated expression of GFP in eosinophils and monocytes of HOM hMRP8-ATTAC mice compared to WT mice (Appendix A), although these levels were considerably lower than the GFP levels of neutrophils (Figure 1J). Dendritic cells and tissue resident macrophages, such as red pulp macrophages in the spleen, Kupffer cells in the liver, alveolar macrophages in the lung, and the CD169^+^ macrophages in the lymph node, were GFP negative (Appendix A). As expected, CD3^+^ T cells also did not express GFP (Appendix A).

We next set out to investigate at which stage of hematopoietic differentiation (Figure 2A) the ATTAC construct was expressed. By flow cytometry analysis of bone marrow cells from HOM hMRP8-ATTAC mice, we observed that Lineage^−^Sca1^+^cKIT^−^ (LSK), common myeloid progenitors (CMP), and the megakaryocyte erythroid progenitor (MEP) were GFP negative (Figure 2B). Along the line of neutrophil differentiation, we observed a modest increase in GFP expression, albeit not significant, in granulocyte monocyte progenitor (GMP), which intensified in promyelocytes and even more in neutrophils (Figure 2B). A small increment in GFP expression was also seen in monocyte dendritic cell progenitor (MDP) and monocytes (Figure 2B), in line with our observation that monocytes from several organs showed modest GFP expression (Appendix A). Altogether, these data indicate that the hMRP8 promoter becomes active in the promyelocytes and MDP populations, inducing transgene expression in these cells and in the following differentiated populations. 

### 3.2. Depletion of Neutrophils in the hMRP8-ATTAC Mouse Model upon Dimerizer Injection

In order to test the depletion of transgene-expressing cells, we injected hMRP8-ATTAC and non-transgene-carrying control WT mice with the dimerizer AP20187 every day for one week (Figure 3A). Two days of dimerizer treatment resulted in a strong reduction of circulating neutrophils in hMRP8-ATTAC mice (Figure 3B,C). A modest temporary decrease in neutrophils was observed in the control groups (in vehicle-treated hMRP8-ATTAC and vehicle/dimerizer-treated WT mice) compared to pre-treatment samples (Figure 3B,C and Appendix A). After a week of dimerizer treatment, neutrophil frequency and counts were strongly reduced in blood and lungs and modestly in spleen and bone marrow of hMRP8-ATTAC mice (Figure 3B–D and Appendix A). The small proportion of neutrophils that remained in the circulation upon dimerizer treatment showed a similar size and granularity as non-depleted neutrophils in vehicle treated mice (Figure 3E). In addition, cell surface expression of CD11b and Ly6G markers were similar between the remaining neutrophils in dimerizer-treated mice and non-depleted neutrophils in vehicle-treated mice (Figure 3F,G), suggesting that the small remaining neutrophil population is not different from depleted neutrophils; thus, dimerizer treatment does not seem to deplete a particular subpopulation of neutrophils. Looking at the other myeloid cell populations, a small, but significant, relative increase in monocytes was observed in the spleen and bone marrow, but not in circulation and lung, of dimerizer-treated hMRP8-ATTAC mice (Appendix A). The frequency of eosinophils was not influenced by the dimerizer treatment, except for a reduction in pulmonary eosinophils (Appendix A), and tissue resident macrophages in the spleen and lung were also not affected (Appendix A). We observed a small increase in circulating neutrophils and monocytes, but not in circulating eosinophils and resident macrophages, in WT mice after dimerizer injection, probably due to a modest inflammatory response induced by the daily injections and/or the compound (Appendix A). Overall, these data illustrate that neutrophils are efficiently depleted in the circulation and lungs and are modestly reduced in spleen and bone marrow upon dimerizer administration. Other immune cell populations were minimally affected by dimerizer treatment.

Next, we assessed whether the hMRP8-ATTAC model was eligible for long-term neutrophil depletion studies. Long-term dimerizer treatment for five weeks in hMRP8-ATTAC mice resulted in a prolonged reduction of circulating neutrophils (Figure 3H,I). Since the transgene was starting to be expressed in the bone marrow in promyelocytes (Figure 2B), we assessed the effect of dimerizer treatment on neutrophils in the bone marrow after the 5-week treatment regimen. However, the neutrophil population in the bone marrow was unaffected by long-term dimerizer treatment, similar to the monocyte population (Appendix A). Whether this is caused by potential lower activity of the hMRP8 promotor in neutrophils in the bone marrow (Figure 2B) versus blood (Figure 1E) or by other mechanisms is unclear. No significant changes in the circulating levels of monocytes and eosinophils were observed over time (Appendix A), confirming that dimerizer treatment—also when administered for a longer period—does not affect other myeloid cells than neutrophils. Together, these findings indicate that the hMRP8-ATTAC mouse model can be used for long-term systemic neutrophil ablation. 

### 3.3. Dimerizer Treatment Is Not Effective for Long-Term Neutrophil Depletion in Tumor-Bearing hMRP8-ATTAC Mice

We and others have reported previously that neutrophils accumulate in several organs and circulations of tumor-bearing mice and that their expansion is proportional to primary tumor size [3,26,27] and to the genetic make-up of the tumor [21,28]. To investigate the depletion efficacy of neutrophils in the hMRP8-ATTAC model in a mammary tumor setting, we orthotopically transplanted mammary tumor pieces, derived from the transgenic *K14-cre;Cdh1^F/F^;Trp53^F/F^* (KEP) mouse mammary tumor model [20,29], in the mammary fat pad of hMRP8-ATTAC mice. Vehicle or dimerizer treatment started at a tumor size of 25 mm^2^ and continued until the tumor size reached the humane endpoint (225 mm^2^) (Figure 4A). As a positive control, we treated a group of tumor-bearing hMRP8-ATTAC mice with the neutralizing antibody against Ly6G (clone 1A8) that temporarily depletes neutrophils in KEP tumor-bearing mice, as we reported previously [27,30]. As expected, the frequency of circulating neutrophils increased during tumor growth in vehicle-treated mice (Figure 4B). Anti-Ly6G effectively reduced neutrophil levels during the first phase of tumor outgrowth, but at the end of the experiment, anti-Ly6G had lost its depletion efficacy, most likely due to the development of neutralizing antibodies against the rat antibody (Figure 4B,C and Appendix A). In contrast, no reduction in neutrophil levels was observed during treatment with the dimerizer (Figure 4B,C). 

We used a second independent transplantation-based mammary tumor model to assess the depletion efficacy of neutrophils in the hMRP8-ATTAC model. We orthotopically transplanted a breast cancer cell line, derived from the *Wap-cre;Cdh1^F/F^;Akt1^E17K^* (WEA) genetically engineered mouse model (GEMM) for mammary tumorigenesis [31], into the mammary fat pad of hMRP8-ATTAC recipients and applied the same dimerizer treatment regimen (Figure 4D).Like in the KEP tumor model, circulating neutrophil levels increased with tumor size (Figure 4E). Dimerizer treatment significantly reduced circulating neutrophil levels during primary tumor growth at a tumor size of 100 mm^2^, although neutrophil levels of dimerizer-treated mice were still increased when compared to the levels at 25 mm^2^. At the end stage, no difference in neutrophil levels between vehicle-treated versus dimerizer-treated mice was observed, indicating that the neutrophil-depleting effect of the dimerizer in tumor-bearing mice was temporary and modest (Figure 4E,F). These data indicate that dimerizer treatment of hMRP8-ATTAC mice was not sufficient to deplete neutrophils during the progression of mammary KEP and WEA tumors. 

## 4. Discussion

As one of the first immune cells that arrive in an inflamed area, neutrophils represent an indispensable component to fight infections. Neutrophils are not only important against pathogens, but they also play a critical role in tumor progression and metastasis formation, having both pro- and anti-tumoral functions [3]. In order to investigate the functional contribution of neutrophils to homeostasis, specific inflammatory stimuli, infections, or cancer, it is desirable to deplete or inhibit these cells and monitor the consequences. Thus far, the most commonly used in vivo strategies to ablate neutrophils are treatment with the depleting anti-Ly6G and anti-Gr1 antibodies and the use of CXCR2 antagonists to interfere with neutrophil trafficking. However, the short-term efficacy of anti-Ly6G, its inefficiency at depleting neutrophils in C57BL/6J mice, and the non-specificity of anti-Gr1 and CXCR2 antagonists are the main concerns. In addition, several genetic models have been generated. For example, the *LysMcre;Mcl-1^F/F^* model is deficient in the anti-apoptotic molecule Mcl-1, a Bcl-2 family member, in neutrophil and monocyte/macrophage populations [32]. This mouse model shows decreased numbers of neutrophils, while macrophage levels were not influenced because of a compensatory upregulation of other Bcl-2 family members [32]. Others have used mice bearing point mutations in or the full knock-out of the growth factor independence 1 (Gfi1), which is important for neutrophil and T cell development [33,34]. In addition, since G-CSF is the main growth factor for neutrophil development, G-CSF^−/−^ mice display neutropenia and deficiency in granulocyte and macrophage progenitors [35]. Although these models are characterized by a decrease in neutrophil levels, they do not allow to conditionally induce or revert neutrophil depletion. Furthermore, other concerns of the aforementioned genetic models are that they may have unknown developmental defects, they may have developed compensatory mechanisms to cope with the absence of neutrophils, or, since neutrophils are critical for immune defense, they may have a different composition of the host microbiome. Another mouse model for neutrophil ablation was developed by intercrossing ROSA-iDTR^KI^ mice, which bear a Cre-inducible simian diphtheria toxin receptor (DTR), with hMRP8-Cre mice to induce DTR expression (*hMRP8cre;ROSA-iDTR^KI^*) in S100A8-expressing cells [11]. It is known, however, that long-term treatment with DT induces an immune reaction against the compound itself, blocking its effect [12,13]. Such an immunogenic response also occurs against anti-Ly6G antibodies, hampering their long-term efficacy. Recently, this immunogenicity has been overcome by a double antibody-based strategy using a combination of anti-Ly6G and anti-rat antibodies to clear neutralizing rat antibodies and improve neutrophil depletion [7].

There is a clear need for genetic models that allow for long-term efficient and specific conditional and reversible neutrophil depletion. Here, we described a novel transgenic mouse model for neutrophil depletion that relies on the activation of the caspase 8-FKBP fusion protein upon injection with a chemical dimerizer. The hMRP8-ATTAC mouse model showed strong hMRP8 promotor activity in neutrophils and a much lower activity in monocytes and eosinophils, as defined by GFP expression in these cells. In the bone marrow, the transgene was expressed at very low levels at the GMP stage and was upregulated during differentiation along the monocytic and granulocytic lineages. Importantly, because S100A8 expression is downregulated during the differentiation of GMPs into macrophages and DCs [36,37], we did not observe GFP expression in these cells. As expected, because of their high transgene expression, neutrophils in the hMRP8-ATTAC model were very sensitive to the effects of the dimerizer, resulting in decreased numbers of these cells in blood, lung, and spleen, and to a lesser extent in the bone marrow. Successful neutrophil depletion was achieved after only two days of dimerizer treatment and lasted at least up to five weeks during long-term dimerizer treatment. Although dimerizer treatment was effective in reducing neutrophils in a homeostatic situation in this mouse model, it was not successful in two independent cancer models. Several reasons might be responsible for the lack of neutrophil depletion in tumor-bearing hMRP8-ATTAC mice. Firstly, it is well established that tumors stimulate granulopoiesis in the bone marrow and release immature and mature neutrophils into the circulations [3,26,27,38]. In these chronic inflammatory conditions, neutrophils are constantly produced in excessive amounts and perhaps exceeding the activity of the dimerizer. Secondly, the half-life of the dimerizer is approximately 5 h (Takara Bio Inc., San Jose, CA, USA), so the daily dimerizer injection schedule that was performed in our experiments might not be enough to suppress the continuous tumor-induced neutrophil expansion. Thirdly, neutrophil longevity is thought to be ~24 h in mice depending on the tissue they reside in [39], and these cells are quickly and continuously replenished [3,40], making it perhaps challenging to maintain a sufficient level of depletion. Indeed, previous studies using the ATTAC system that aimed at depleting adipocytes, podocytes, pancreatic β-cells, and p16^Ink4a^-positive senescent cells showed a pronounced and long-lasting depletion of these cell populations, although these studies used even lower concentrations of the dimerizer per injection compared to our study [14,15,16,17,41]. These cell types are not renewed at the same rate as neutrophils, likely explaining the more stable depletion in these models. More frequent dimerizer administration for the tumor-bearing hMRP8-ATTAC model might be a solution, or future studies should aim at increasing the dimerizer half-life or at developing a formulation of the dimerizer that remains stable in the animals’ chow. Fortunately, the chemical dimerizer is well tolerated, and no signs of toxicity or immunogenicity against the dimerizer were observed during the long-term dimerizer treatment of five weeks. Moreover, we did not observe any compensatory mechanisms in the hMRP8-ATTAC model by means of increased production of neutrophils during long-term dimerizer treatment or enhanced production of neutrophils in the bone marrow upon dimerizer treatment. In addition, others have performed long dimerizer treatment for four weeks [14] and even for four months [41] in immunocompetent mice, resulting in efficacious target cell depletion, suggesting that the dimerizer does not elicit an immune response, which is an important advantage of the ATTAC strategy. Together, this renders hMRP8-ATTAC mice an eligible model for neutrophil ablation in homeostatic conditions with a low risk of immunogenicity or toxicity. This model can potentially be applied to other disease settings, such as inflammatory diseases, yet it needs to be considered that such inflammatory disease states have not been tested in hMRP8-ATTAC mice so far. In addition, further optimization of the dimerizer administration strategy may be needed when depletion studies in hMRP8-ATTAC mice will be performed at a different life-phase of the mice or when the model will be backcrossed to another genetic background. 

Both cancer and inflammatory conditions can induce S100A8 upregulation in other cell types than neutrophils, potentially interfering as targets for dimerizer treatment. For instance, tumor-derived VEGF-A, TGF-β, and TNF-α may induce S100A8 expression in macrophages and endothelial cells [42], or monocyte/macrophage-derived factors may stimulate its expression in cancer cells [43]. However, in tumor-bearing hMRP8-ATTAC mice, we did not observe loss of tumor-associated macrophages or endothelial cells upon dimerizer treatment (data not shown). 

A potential secondary consequence of dimerizer-mediated neutrophil depletion is the induction of an inflammatory response. For instance, the release of NETs and cytokines or the initiation of NETosis could serve as risks for the induction of an immune response, thus forming a potential adverse event of dimerizer-mediated neutrophil depletion in the hMRP8-ATTAC model. However, the activation of NETosis is unexpected in our model, since NETosis is considered a different type of cell death than caspase-mediated apoptosis, the latter being induced in hMRP8-ATTAC mice. In addition, under physiological circumstances, neutrophils become non-functional when apoptosis is initiated and are subsequently cleared via phagocytosis [44,45]. In this regard, it is unlikely that such an inflammatory response would occur in the hMRP8-ATTAC model, since the neutrophil release of inflammatory mediators is limited during apoptosis. In our study, no signs of inflammation were observed in terms of animal behavior, macroscopic inspection, or comprehensive immune characterization (Figure 3 and Appendix A).

Differently from *hMRP8cre;ROSA-iDTR^KI^* mice, which show a small reduction in monocytes upon DT administration [11], monocyte proportions modestly increased in several tissues of non-tumor-bearing hMRP8-ATTAC and WT mice and tumor-bearing hMRP8-ATTAC mice upon dimerizer injection. Interestingly, a similar increase in monocytes was also observed in the Gfi1 mutated and knock-out models [33,34]. Most likely, this is a relative increase in monocytes due to neutrophil reduction or an increment in their absolute number due to dimerizer-mediated stimulation of myelopoiesis. The latter is supported by the observation of a higher proportion of monocytes in the bone marrow of dimerizer-treated hMRP8-ATTAC mice. Although the transgene is expressed at low levels by monocytes, dimerizer administration does not induce monocyte apoptosis in hMRP8-ATTAC mice, confirming high specificity of dimerizer-induced neutrophil targeting in this model.

In conclusion, we here described the generation and characterization of a new mouse model for the long-term conditional ablation of neutrophils. Although successful conditional depletion of neutrophils through the activation of caspase 8 was achieved in hMRP8-ATTAC mice, future studies should focus on improving the model in order to sustain neutrophil depletion in a chronically inflamed context. 

## Figures and Tables

**Figure 1 cells-11-02346-f001:**
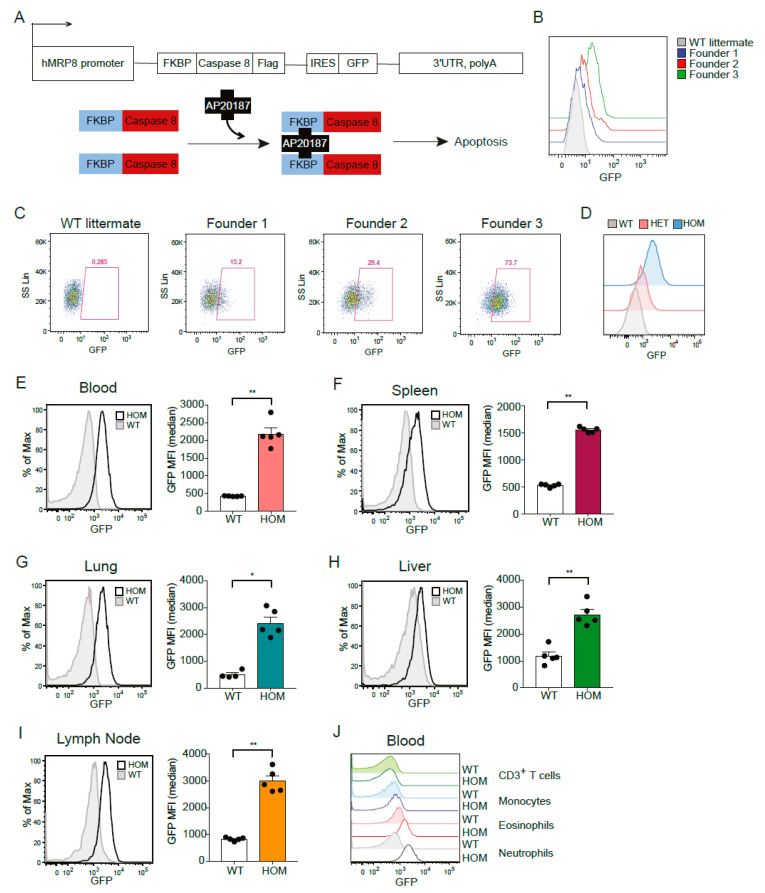
Generation and characterization of hMRP8-ATTAC mice. (**A**) Schematic representation of the ATTAC construct and the mechanism of apoptosis induced by the dimerizer AP20187. (**B**,**C**) Histograms and dot plots showing GFP expression in circulating neutrophils of four heterozygous hMRP8-ATTAC founders, as determined by flow cytometry. One of the founders does not carry the transgene as detected by PCR (WT littermate); the other three carry the transgene and express GFP at different levels (Founders 1, 2, 3). (**D**) Representative histograms of GFP expression in circulating neutrophils of heterozygous and homozygous hMRP8-ATTAC mice compared to WT littermates, as determined by flow cytometry. (**E**–**I**) Representative histograms of GFP expression and quantification of the Median Fluorescence Intensity (MFI) of GFP in neutrophils from blood (**E**), spleen (**F**), lung (**G**), liver (**H**), and lymph nodes (**I**) of homozygous hMRP8-ATTAC mice (*n* = 5) compared to WT littermates (*n* = 4–5), as determined by flow cytometry. (**J**) Representative histogram comparing GFP expression levels in circulating neutrophils, eosinophils, monocytes, and CD3^+^ T cells of homozygous hMRP8-ATTAC mice and WT littermates. WT, wild type; HET, heterozygous; HOM, homozygous. Data presented in E-I are mean values ± SEM. * *p* < 0.05, ** *p* < 0.01 by Mann–Whitney test.

**Figure 2 cells-11-02346-f002:**
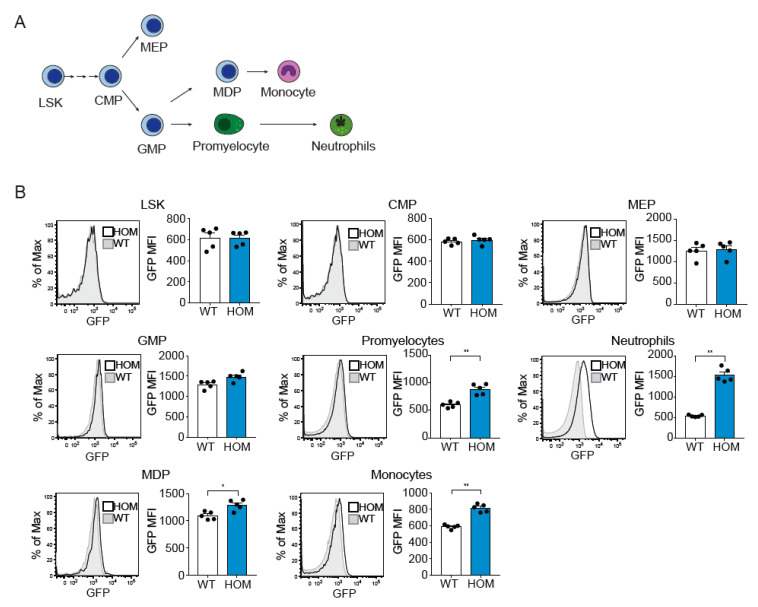
The ATTAC transgene in hMRP8-ATTAC mice starts to be expressed at the GMP level in the bone marrow. (**A**) Schematic representation of the hematopoietic tree indicating the different steps of myelopoiesis. (**B**) Representative histograms of GFP expression and quantification of the Median Fluorescence Intensity (MFI) of GFP in different hematopoietic cell populations depicted in (**A**) the bone marrow of homozygous female hMRP8-ATTAC mice and WT littermates (*n* = 5/group), as determined by flow cytometry. WT, wild type; HOM, homozygous. LSK, Lineage^-^SCA1^+^Kit^-^; CMP, common myeloid progenitor; MEP, megakaryocyte-erythroid progenitor; GMP, granulocyte-monocyte progenitor; MDP, monocyte-dendritic cell progenitor. Data presented in B are mean values ± SEM. * *p* < 0.05, ** *p* < 0.01 by Mann–Whitney test.

**Figure 3 cells-11-02346-f003:**
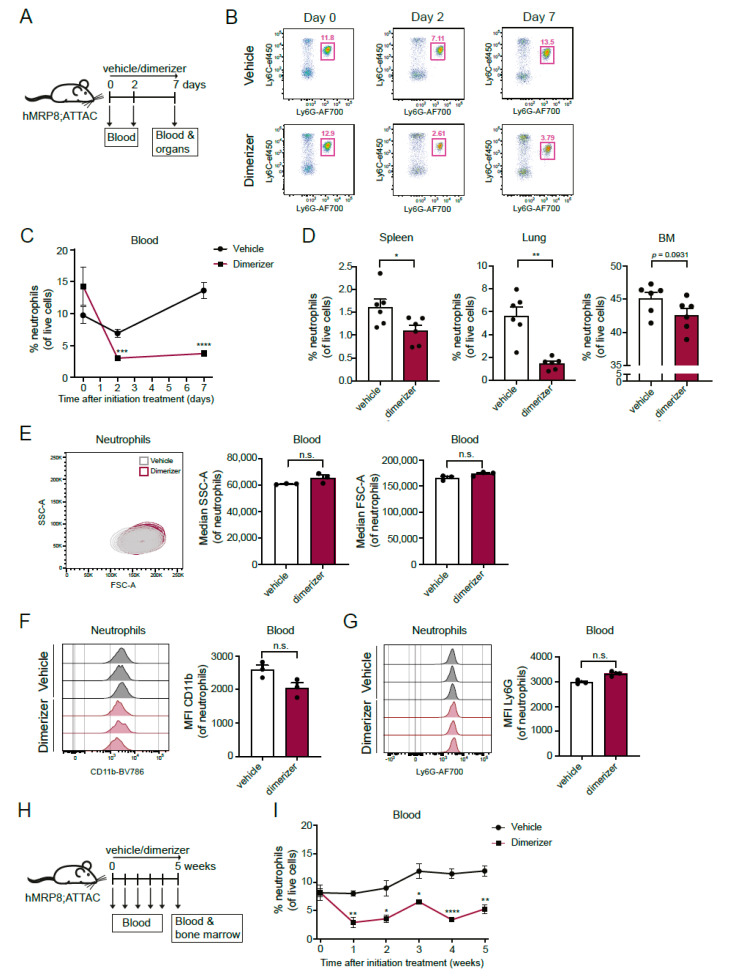
Dimerizer treatment reduces neutrophil frequency in hMRP8-ATTAC mice. (**A**) Schematic representation of the experimental design. Female homozygous hMRP8-ATTAC mice were treated daily with vehicle or dimerizer for seven consecutive days. Blood was analyzed by flow cytometry before the start of the treatment and two days after the start of the treatment. At the seventh day, mice were sacrificed and tissues were processed and analyzed by flow cytometry. (**B**) Representative dot plots showing neutrophils (as percentages of live cells) in the blood of either vehicle- or dimerizer-treated hMRP8-ATTAC mice before treatment (day 0) and after two or seven days of treatment. (**C**) Neutrophil frequencies in the blood of vehicle- or dimerizer-treated hMRP8-ATTAC mice (*n* = 6/group) over a period of seven days. (**D**) Frequencies of neutrophils in spleen, lung and bone marrow of hMRP8-ATTAC mice (*n* = 6/group) treated with vehicle or dimerizer for seven consecutive days. (**E**–**G**) Characterization of remaining neutrophils upon seven days of vehicle or dimerizer treatment showing a dot plot and quantification of their granularity (SSC) and size (FSC) (**E**), and histograms and quantification of the Median Fluorescence Intensity (MFI) of cell surface markers CD11b (**F**) and Ly6G (**G**). (**H**) Experimental design of long-term dimerizer administration. Female homozygous hMRP8-ATTAC mice were daily treated with vehicle or dimerizer for five weeks. Blood was drawn once a week, starting at day 0 before treatment initiation. After five weeks of treatment, mice were sacrificed, and blood and bone marrow were analyzed using flow cytometry. (**I**) Circulating neutrophil frequencies were determined by flow cytometry during five weeks of vehicle or dimerizer treatment in hMRP8-ATTAC mice (*n* = 6/group). Data presented in (**C**–**G**) and (**I**) are mean values ± SEM. * *p* < 0.05, ** *p* < 0.01, *** *p* < 0.001, **** *p* < 0.0001 by unpaired *t*-test followed by Holm-Sidak’s correction for multiple comparisons (**C**,**I**) or * *p* < 0.05, ** *p* < 0.01 by Mann–Whitney test (**D**–**G**). n.s., not significant; MFI, Median Fluorescence Intensity.

**Figure 4 cells-11-02346-f004:**
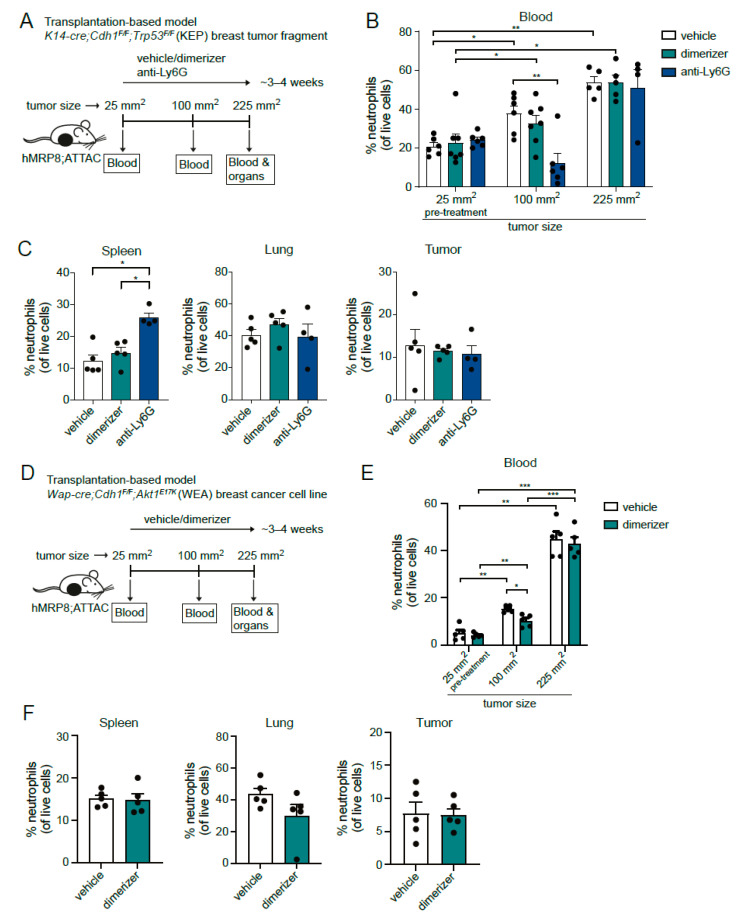
Dimerizer treatment is ineffective in depleting neutrophils on the long-term in mammary tumor-bearing hMRP8-ATTAC mice. (**A**) Illustration of the experimental design. Female homozygous hMRP8-ATTAC mice were orthotopically transplanted in the mammary fat pad with *K14-cre;Cdh1^F/F^;Trp53^F/F^* tumor fragments. Mice were treated with either vehicle, dimerizer or anti-Ly6G until sacrifice at a tumor size of 225 mm^2^. Blood was collected before the start of the treatment (25 mm^2^) and at a tumor size of 100 mm^2^. At a tumor size of 225 mm^2^, mice were sacrificed, and blood and organs were collected for flow cytometry analysis. (**B**) Frequency of neutrophils in the blood of hMRP8-ATTAC mice treated as indicated (*n* = 6–7/treatment group) at a tumor size of 25 mm^2^, 100 mm^2^, and 225 mm^2^, as determined by flow cytometry. (**C**) Frequency of neutrophils in spleen, lung, and tumor of endpoint hMRP8-ATTAC mice (*n* = 4–5), as determined by flow cytometry. (**D**) Schematic of the experimental set-up. Orthotopic intramammary injection of *Wap-cre;Cdh1^F/F^;Akt1^E17K^* (WEA) breast cancer cells was performed in female homozygous hMRP8-ATTAC mice. Blood was collected before the start of the treatment (25 mm^2^) and at a tumor size of 100 mm^2^. Mice were treated with either vehicle or dimerizer until the day of sacrifice, when the tumor size reached 225 mm^2^. After sacrifice, blood and organs were collected for flow cytometry analysis. (**E**) Flow cytometry analysis of blood showing circulating neutrophils of mice treated as indicated at a tumor size of 25 mm^2^, 100 mm^2^, and 225 mm^2^ (*n* = 5/treatment group). (**F**) Neutrophil frequency of WEA tumor-bearing hMRP8-ATTAC mice (*n* = 5/treatment group) at the endpoint of the experiment (tumor size of 225 mm^2^) in spleen, lung, and tumor, as determined by flow cytometry. Data presented in B and E are mean values ± SEM. * *p* < 0.05, ** *p* < 0.01, *** *p* < 0.001 by mixed-effects analysis (**B**) or two-way ANOVA (**E**), both with Sidak’s multiple comparisons test. Data presented in C and F are mean values ± SEM. * *p* < 0.05 by Mann–Whitney test.

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
