# Peer review of "hMRP8-ATTAC Mice: A New Model for Conditional and Reversible Neutrophil Ablation"

_cells, 2022, doi:10.3390/cells11152346_

Round 1
Reviewer 1 Report
In the manuscript, “hMRP8-ATTAC Mice: A new Model for Conditional and Reversible Neutrophil Ablation,” the authors describe a novel mouse to ablate neutrophils in vivo. The authors demonstrate depletion of neutrophil in several organs at baseline using their model. Additionally, they demonstrate that depletion begins with GMP in the marrow, and is largely exclusive to neutrophils. Finally this depletion can be sustained for several weeks with repeated treatment with their dimerizer. Unfortunately, neutrophil depletion does not occur in either of the tumor models the authors presented (despite at least transient depletion using anti-Ly6G antibodies). This makes the utility of this model unclear. The authors mention that additional optimization needs to occur in the tumor models, I would suggest that the current publication of this model is somewhat premature until the authors have performed this optimization. Alternately, the authors could explore other neutrophil-based diseases where this mouse model could be more useful, for example acute infection, neutrophil-predominant autoimmune/inflammatory diseases (RA, IBD).
Additional comments/suggestions below:
The mechanism of depletion of neutrophils is through caspase-mediated apoptosis. Are there any systemic effects (ie inflammatory) of apoptosis of such a large number of cells?
The authors should consider a head-to-head comparison of neutrophil depletion in their model compared to the currently existing antibody methods. They do this for the tumor models, but it would also be interesting to see in at steady state as well. Also, the depletion data is expressed as % of live cells, it would also be helpful to see absolute number of neutrophils in these experiments.
Is there an increase in infectious complications in the mice undergoing several weeks of neutrophil depletion?
The mice used in this experiment are on the older side, (17-19 weeks and 17-23 weeks) is there a particular reason for this choice?
Reviewer 2 Report
Duits et al presents a novel approach to ablation of neutrophils in reversible form. Since the neutrophils are key players in innate immune response, the discussed area seems to be emerging and truly interesting to the Readers.
The design of the experiment, in terms of mice generation, has no weak points and has been designed accordingly to the state-of-the-art. I would recommend two-way confirmation of the purity of generated pups, but for basic science paper, the evidence is good.
What brings me some thoughts is the results section that shows that the numbers of neutrophils were lower, but still present. I came to an idea that the proposed way of ablation might be relevant to only particular subtype of neutrophils - e.g. in terms of maturity or only for activated ones. It might be so inetersting to check this issue. Moreover, it would be of great interest whether ablation leads to increased activity of remaining in the bloodstream neutrophils? Measuring the levels of NETs markers (citH3, NE or exDNA) will be appreciated.
Moreover, I understand that this approach is relevant to many diseases, but I would suggest shortening the introduction - it is a bit too long and introduces lots of irrelevant background.
Moving further, I am very surprised by using relatively very old mice. Since neutrophils biology changes across the lifespan, younger mice should be use to validate the findings.
Also please increase the quality of Figure 1 - it is blurry and fuzzy.
Please use the same font across the whole draft. Lines 29, 100 and more.
What is the safety of proposed model? How high is the risk that compensatory mechanisms will introduce lots of bias to the further studies?
How real is the risk that ablated neutrophils become releasing exDNA or formed NETs?
In general, very solid paper, great merit as well as novel approaches.
Round 2
Reviewer 1 Report
All of my scientific concerns have been adequately addressed by the authors.
Reviewer 2 Report
The Authors in a very clear and kind manner addressed all my major and minor comments as well as introduced some minor changes in the manuscript body.
However, I do not feel that the Authors took into account performing additional experiments to support their conclusions in terms of mice age and assessment of neutrophil activation. I am aware that using very old mice is related to the study type of cancer, however, the proposed model is described as an ablation model that might be used for various type of studies.
So, my final recommendation is:
The Authors should choose one of the following;
- perform the same set of murine experiments using 12-14-week old mice to confirm the "universal" approach for neutrophil ablation in different biological conditions.
- or just tone down the potential usage of this model and point out that this model might work solely for conditions that have been tested. Even the title of the paper suggests the introduction of a widely adapted model with the potential to use in many different sets and settings. This has not been supported by the results so far.
This model needs more evidence and more testing to be considered as a global model. Please, do not take this personally, I try to avoid the situation when other folks will give you a hard time since your model will not work in their murine models of e.g. psoriasis, LPS-induced lung injury, or hemophilic arthropathy.
Anyway, looking for the revised version.
